# Lack of Association between Antimicrobial Consumption and Antimicrobial Resistance in a HIV Preexposure Prophylaxis Population: A Cross-Sectional Study

**DOI:** 10.3390/antibiotics13020188

**Published:** 2024-02-16

**Authors:** Thibaut Vanbaelen, Jolein Laumen, Christophe Van Dijck, Tessa De Block, Sheeba Santhini Manoharan-Basil, Chris Kenyon

**Affiliations:** 1STI Unit, Department of Clinical Sciences, Institute of Tropical Medicine, 2000 Antwerp, Belgium; tvanbaelen@itg.be (T.V.); jolein.laumen@sciensano.be (J.L.); cvandijck@itg.be (C.V.D.); sbasil@itg.be (S.S.M.-B.); 2Clinical Reference Laboratory, Department of Clinical Sciences, Institute of Tropical Medicine, 2000 Antwerp, Belgium; tdeblock@itg.be; 3Division of Infectious Diseases and HIV Medicine, University of Cape Town, Cape Town 7700, South Africa

**Keywords:** antimicrobial resistance, PrEP, saturation, *Neisseria gonorrhoeae*, *Neisseria subflava*, gonorrhoea, stewardship

## Abstract

Background: In antibiotic naïve populations, there is a strong association between the use of an antimicrobial and resistance to this antimicrobial. Less evidence is available as to whether this relationship is weakened in populations highly exposed to antimicrobials. Individuals taking HIV preexposure prophylaxis (PrEP) have a high intake of antimicrobials. We previously found that there was no difference in the prevalence of pheno- and genotypic antimicrobial resistance between two groups of PrEP clients who had, and had not, taken antimicrobials in the prior 6 months. Both groups did, however, have a higher prevalence of resistance than a sample of the general population. Methods: In the current study, we used zero-inflated negative binomial regression models to evaluate if there was an individual level association between the consumption of antimicrobials and 1. the minimum inhibitory susceptibilities of oral *Neisseria subflava* and 2. the abundance of antimicrobial resistance genes in the oropharynges of these individuals. Results: We found no evidence of an association between the consumption of antimicrobials and the minimum inhibitory susceptibilities of oral *Neisseria subflava* or the abundance of antimicrobial resistance genes in these individuals. Conclusions: We conclude that in high-antimicrobial-consumption populations, the association between antimicrobial consumption and resistance may be attenuated. This conclusion would not apply to lower-consumption populations.

## 1. Introduction

A large body of evidence has established incontrovertibly that antimicrobial exposure can result in antimicrobial resistance (AMR) [1,2,3]. In a similar vein, populations with more extensive exposure to an antimicrobial typically have a higher prevalence of resistance to that antimicrobial [3,4,5,6,7,8]. What is less clear is if this effect is saturated at extremely high levels of exposure. In other words, is there an antimicrobial exposure threshold above which the association between consumption and resistance is attenuated?

In the case of macrolides, a number of ecological level studies have found evidence of saturation. For example, once the population level consumption of macrolides increases to approximately 700 defined daily doses per 1000 population per year, the prevalence of macrolide resistance in *Treponema pallidum* increases rapidly from less than 10% to over 90% with no evidence of a further increase [9]. Similar positive associations, although with not as clear evidence for saturation, have been described for macrolides in other species such as *Streptococcus pneumoniae*, *Mycoplasma genitalium* and *Helicobacter pylori* [9,10,11,12,13,14,15,16]. 

It is plausible that the individual-level association between macrolide consumption and resistance may be lost in very high consumption populations for a number of reasons, as outlined in Figure 1. Such a saturation effect could have important consequences as it may mean that one may miss the association between antimicrobial use (AMU) and AMR in high-consumption populations. This question is particularly prescient in the current era, where a number of countries are considering the roll out of doxycycline post exposure prophylaxis in men who have sex with men (MSM) to reduce the incidence of certain STIs [17,18]. This decision has been based on randomised controlled trials showing evidence of reduced STI incidence and little or no risk of AMR emerging [17,18]. These trials have, however, been performed with MSM in HIV PrEP cohorts with a high consumption of antimicrobials. 

In previous studies, we have found that antimicrobial consumption in PrEP cohorts exceeds resistance-inducing thresholds for various bacteria–antimicrobial combinations by up to seven folds [19,20]. We have also found evidence compatible with antimicrobial saturation in these cohorts. For example, in one study, we found higher azithromycin, ciprofloxacin and ceftriaxone minimum inhibitory concentrations (MICs) of oral commensal *Neisseria* species in MSM on PrEP than the Belgian general population, but no difference between MSM who had and who had not taken antimicrobials in the past 6 months [21]. We also found that MSM on PrEP had a higher abundance of genes conferring resistance to macrolides, beta-lactams and fluoroquinolones than the general population. However, once again, we found no difference between MSM who had and who had not taken antimicrobials in the prior 6 months [22]. 

The associations between AMU and AMR in these three groups of individuals were tested only at the group level, which may obscure individual-level associations between the consumption of a particular antimicrobial and resistance to that antimicrobial. The objective of this paper was, therefore, to conduct a secondary data analysis of these three groups to assess if there is an individual-level association between the consumption of four classes of antimicrobials (macrolides, fluoroquinolones, tetracyclines and beta-lactams) and geno- and phenotypic markers of AMR. For the assessment of phenotypic resistance, we focused on *Neisseria subflava*. *N. subflava* is the most prevalent species of commensal *Neisseria* spp. to inhabit the human oropharynx [23,24]. By virtue of their higher prevalence (they are detected in close to 100% of humans) than the pathogenic *Neisseria* spp.; commensal *Neisseria* spp. such as *N. subflava* are more susceptible to AMR-inducing effects of high antimicrobial exposure [23,24,25]. Furthermore, AMR in the pathogenic *Neisseria* spp. frequently emerges through the horizontal gene transfer of resistance-conferring DNA from the commensal *Neisseria* spp. [25,26]. As such, commensal Neisseria have been proposed to be a useful early warning system of excessive antimicrobial exposure [23,24,25]. 

## 2. Methods

### 2.1. Study Population 

The study population consisted of a cross sectional sample of three groups of 32 individuals: one group from the general population and two groups of MSM attending the HIV preexposure prophylaxis (PrEP) clinic of the Institute of Tropical Medicine (ITM), Antwerp. In total, 64 MSM were included at the baseline visit of the PReGo study in 2019–2020. PReGo was a randomised placebo controlled trial that assessed the efficacy of an antiseptic mouthwash to prevent STIs among higher-risk MSM [28]. One of the inclusion criteria for PReGo was a documented infection with gonorrhoea, chlamydia or syphilis in the preceding 24 months. Thus, they had consumed at least one course of antimicrobial in the preceding 2 years. The first PReGo participants were enrolled into two groups, depending on if they had (group 1) or had not (group 2) consumed any antimicrobials in the preceding 6 months. Group 3 (general population) was comprised of ITM employees (male or female) who had not used antimicrobials in the preceding 6 months. This group was recruited by posters in June 2020. The first 32 eligible employees who approached the study team were included in this survey.

### 2.2. Data and Samples 

In brief, an oropharyngeal swab was taken by applying a dry regular flocked swab (COPAN, Brescia, Italy) to both tonsillar pillars and the posterior oropharynx. These were then placed in a cooled transport box and processed as detailed elsewhere [21]. 

### 2.3. DNA Extraction, Shotgun Metagenomic Sequencing and Resistome Characterisation

The methods used for DNA extraction, shotgun metagenomic sequencing and resistome characterisation have been described elsewhere [22]. In brief, metagenomic DNA was extracted from the swabs using the FastDNA™ SPIN Kit (MP Biomedicals, Irvine, CA, USA), followed by the preparation of libraries using the Nextera XT DNA library preparation kit (Illumina Inc., San Diego, CA, USA) and sequencing using 2 × 250 bp Miseq and 2 × 150 bp NextSeq500 (Illumina Inc., USA). 

### 2.4. Taxonomic and Resistome Characterisation 

After the read trimming and filtering of low-quality reads using trimmomatic (v0.39), human reads were removed by mapping the reads against the human reference genome (GRCh38, accession GCF_000001405.26) using Burrows–Wheeler Alignment with default parameters [29,30]. Samples with less than 5500 non-human reads were discarded in order to avoid issues relating to variations in sequencing depth. The abundance of antimicrobial resistance genes (ARGs) was estimated using the MEGARes 2.0 database and BWA-MEM with default settings [31]. ResistomeAnalyzer (https://github.com/cdeanj/resistomeanalyzer, accessed 23 October 2023) was used to classify ARGs with a gene fraction greater than 80% into types, classes, and gene groups for further analyses. ARGs were normalised by the number of bacterial reads per sample (as estimated by Bracken) and multiplied by 10^6^ in order to obtain reads per million (RPM). Single nucleotide polymorphisms were not considered for further analysis [32,33].

### 2.5. Culture and Antimicrobial Susceptibility Determination of Neisseria Species

The ESwabs™ were inoculated onto Modified Thayer-Martin Agar and incubated at 37 °C and 5% carbon dioxide. After 48 h of incubation, Gram-negative, oxidase-positive colonies were selected, enriched and stored in skim milk at −80 °C.

Isolates were identified to the species level via Matrix-Assisted Laser Desorption/Ionization-Time-of-Flight mass spectrometry (MALDI-TOF MS), on a MALDI Biotyper^®^ Sirius IVD system using the MBT Compass IVD software V 1.1and library (Bruker Daltonics, Bremen, Germany). For further details, please see Laumen et al. [21]. Isolates identified as *N. perflava* and *N. flavescens* were grouped into one category with *N. subflava* [34]. 

The minimum inhibitory concentrations (MICs) of *Neisseria* species to azithromycin, ceftriaxone and ciprofloxacin were determined on GC agar plates using ETEST^®^ (bioMérieux Marcy-l’Étoile, France) incubated for 24 h at 37 °C and 5% CO_2_.

*N. subflava* was detected in 58/96 (60.4%) individuals. The other *Neisseria* species were detected in less than 15% of individuals and therefore not used for further analyses (*N. mucosa* (14/96, 14.6%), *N. oralis* (8/96, 8.3%), *N. cinerea* (3/96, 3.1%), *N. elongata* (3/96, 3.1%), *N. lactamica* (2/96, 2.1%) and *N. bacilliformis* (1/96, 1.0%)). *N. subflava* has a number of colony morphologies, and thus, in a number of individuals, more than one colony of this species was characterised. For these individuals, we used the median MICs for all isolates for further analyses.

### 2.6. Ethics 

This study was approved by ITM’s Institutional Review Board (1276/18 and 1351/20) and the Ethics Committee of the University of Antwerp (19/06/058 and AB/ac/003). 

## 3. Data Analysis

### 3.1. Antimicrobial Resistance Gene (ARG) Abundance

Because of the large number of samples with zero-reads for each class of antimicrobials, we used zero-inflated negative binomial regression models to assess associations between ARG abundance and antimicrobial exposure. The outcome variable was the total count of ARGs per class of antimicrobial, per sample. The study group (coded as a categorical variable—1, 2 or 3) and the logarithm of the count of bacterial reads were included as explanatory and offset variables, respectively. 

### 3.2. Antimicrobial Susceptibility of N. subflava

Linear regression was used to assess the association between the azithromycin, ciprofloxacin and ceftriaxone MICs of *N. subflava* (log-transformed) and the consumption of macrolides, fluoroquinolones or ceftriaxone (binary variable), adjusting for the study group (coded 1, 2 or 3). 

For both these analyses, sensitivity analyses were performed, controlling for time since the last ingestion of the relevant antimicrobial (days) and limiting the study to the two MSM groups.

Statistical analyses and data visualisations were conducted using Stata MP V16.1. A *p*-value < 0.05 was considered statistically significant.

## 4. Results

All the PrEP participants and 10 of the general population (31.3%) were men. In the 32 PrEP participants who had used antimicrobials in the prior 6 months, 25 (78.1%), 19 (59.4%), 8 (25%) and 2 (6.3%) had used cephalosporins, macrolides, tetracyclines, or fluoroquinolones, respectively (Table 1).

### 4.1. Antimicrobial Susceptibility of N. subflava

In the 58 individuals with *N. subflava* identified, the median azithromycin MIC was 2.9 µg/mL (IQR 1.6–4; range 0.125–>256 µg/mL), the median ceftriaxone MIC was 0.032 µg/mL (IQR 0.023–0.05; range 0.008–0.5 µg/mL) and the median ciprofloxacin MIC was 0.032 µg/mL (IQR 0.01–0.198; range 0.003–0.875 µg/mL; Table 1). Using the EUCAST’s definitions of AMR for *N. gonorrhoeae*, the prevalences of resistance to azithromycin, ceftriaxone, and ciprofloxacin were 48/58 (82.8%), 4/58 (6.9%) and 24/58 (41.4%), respectively. 

### 4.2. Abundance of Resistance Associated Genes

The median abundances of Macrolides, lincosamides and streptogramines (MLS), beta-lactam, fluoroquinolone and tetracycline resistance-associated genes were 43.6 RPM (IQR 0–293.9), 0 RPM (IQR 0–39.3), 0 RPM (IQR 0–69.7) and 0 RPM (IQR 0–204.6), respectively (Table 1). 

### 4.3. Association AMU/Susceptibility in N. subflava

No association was found between the consumption of macrolides and azithromycin MICs for *N. subflava* (Table 2). The same was true for beta-lactam consumption and ceftriaxone MICs. Both the individuals who had consumed fluoroquinolones did not have *N. subflava* cultured from their oropharynx and the association could therefore not be assessed.

### 4.4. Association AMU/Gene Abundance

There was no association between the consumption of macrolides, beta-lactams, fluoroquinolones or tetracyclines and the abundance of resistance-associated genes to the corresponding class of antimicrobials (Table 3).

The following variables were controlled for in these analyses: study group (1, 2 or 3) and the abundance of bacterial reads. 

Sensitivity analyses, including the time since ingestion of antimicrobials, restricted to the two groups of MSM, produced similar results for both ARG abundance and antimicrobial susceptibility (Appendix A).

## 5. Discussion

We found no association between the use of antimicrobials in the prior 6 months and geno- and phenotypic markers of resistance in our study population. There are a number of possible explanations for our findings. 

Firstly, methodological issues may have obscured an association between antimicrobial consumption and AMR. Our sample sizes were small, and our look-back period on antimicrobial use (AMU) was only 6 months. Larger sample sizes and longer look-back periods may have revealed a positive association. A longer look-back period may be particularly important for antimicrobials such as azithromycin, whose effects on pheno- and genotypic resistance can last for up to 6 months [35,36,37]. The abundance of bacteria and hence ARGs is higher in the gut than in the oropharynx. Although studies have typically found that the effect of antimicrobials on the resistome is similar in these two sites [36,38], it is possible that our results may have differed if we included the gut resistome. In a similar vein, we included only one species, *Neisseria subflava*, as our indicator organism for phenotypic resistance. Furthermore, we used the median MICs of the one to three *N. subflava* colonies isolated per individual. Alternative strategies would include testing a broader range of bacterial species or measuring the proportion of each type of bacteria resistant to each antimicrobial. To characterise the azithromycin susceptibility of commensal *Neisseria* spp., this method could use selective agar plates with 2 µg/mL of azithromycin and without azithromycin to provide a proportion of all commensal *Neisseria* spp. with resistance at this threshold. By assessing the antimicrobial susceptibilities of closer to 100 colonies than 1 colony, this method may be more likely to provide a more representative view of the susceptibility of the target bacteria. In addition, our sequencing depth may not have been deep enough to provide an optimal resolution of bacterial reads. This limitation could have been influenced by the abundance of human reads in samples. Additionally, the exclusion of SNP confirmation or the insufficiency of our ARG bioinformatic pipeline may have hindered the ability to accurately detect differences in ARGs. As a secondary analysis, we assessed for associations between AMU and relevant individual resistance genes but found no associations. Future studies could improve on our methodology by including larger sample sizes, including the sequencing of the gut microbiome with broader range of bacterial species and a more detailed characterisation of antimicrobial exposure that includes the dose and duration of use. 

Secondly, there may be no association between AMU and AMR in this population. As mentioned above, a number of studies have found that our PrEP cohort is exposed to high levels of antimicrobials with consequently high levels of AMR [19,20]. Previous analyses found that both pheno- and genotypic resistance were more prevalent in the two populations of MSM than in the general population, but there was no difference between the two groups of MSM [21,22]. In a recent randomised controlled trial on ceftriaxone versus ceftriaxone plus azithromycin 2 g PO for infection with *N. gonorrhoeae* in this same PrEP cohort, we found that the additional azithromycin had no detectable effect on the abundance of macrolide resistance-genes in the gut or the prevalence of azithromycin resistant oral streptococci or commensal *Neisseria* spp. [39]. A plausible explanation for these findings was the high levels of macrolide resistance found in the baseline specimens—100%/92.5% had azithromycin resistant streptococci/*Neisseria* spp. 

These findings would imply that a degree of AMR saturation has been attained in this population, whereby additional antimicrobial exposure has little detectable effect on AMR. If this were to be the case, this would be an important finding for a number of reasons. Firstly, it would mean we need to be careful about using these populations to evaluate the relationship between AMU and AMR. A contemporaneous example here would be the studies that have concluded that doxycycline PEP is safe as it does not result in a very appreciable AMR in MSM PrEP cohorts [17,18]. Secondly, it suggests the need for enhanced antimicrobial stewardship in these populations to bring AMU down to safer limits. Of note, excessive AMU has been linked not only to AMR but also to reductions in microbial diversity with the associated adverse clinical outcomes [21,22,40]. 

The association between AMU and AMR is complex, with pathways operating at population, individual human and lower levels [27,41]. One of these pathways is the transmission of resistance between individuals, particularly in populations with dense sexual networks, such as PrEP cohorts [42]. Another pathway is the cross-selection of resistance. Baquero et al., for example, have found evidence that the consumption of macrolides is a more important driver of penicillin resistance in *Streptococcus pneumoniae* than beta-lactam consumption [43]. We were unable to evaluate these factors. 

Despite the numerous limitations of our study, our findings serve as a reminder to include populations with low AMUs in studies evaluating the risk of AMR in interventions involving antimicrobials. As noted above, a number of countries are considering the roll out of doxycycline PEP in MSM to reduce the incidence of certain STIs [17,18,44]. The most recent doxycycline PEP RCTs performed in MSM have found little or no risk of AMR emerging [17,18]. These trials have, however, been performed in MSM in HIV PrEP cohorts with a high consumption of antimicrobials. It may be prudent to repeat these studies in populations with lower consumptions of antimicrobials before concluding that the risk of inducing AMR is low. Of note, in this regard, is an RCT of minocycline PEP from 1979 in a group of sailors in the US navy on shore leave. This population presumably had a low antimicrobial consumption, and unlike the more recent studies on cohorts of MSM, this study found that the use of minocycline was associated with selecting for tetracycline resistance in *N. gonorrhoeae* [45]. 

If other studies were to confirm the existence of such a saturation effect, then it would be useful for assessing the optimal methods to measure the effects of excessive antimicrobial exposure and establish safe thresholds of antimicrobial exposure. These could be measured at individual and population levels. One option would be to perform surveillance of oral commensal *Neisseria* and Streptococcal species in key populations such as those taking PrEP [23,46,47]. Once resistance exceeds particular thresholds, antimicrobial stewardship interventions could then be introduced. Finally, it is important to stress that our findings do not imply that excess antimicrobial consumption does not result in AMR. In particular, the findings of this study should not be extrapolated to low antimicrobial consumption populations where AMU has been clearly linked to AMR [3]. 

## Figures and Tables

**Figure 1 antibiotics-13-00188-f001:**
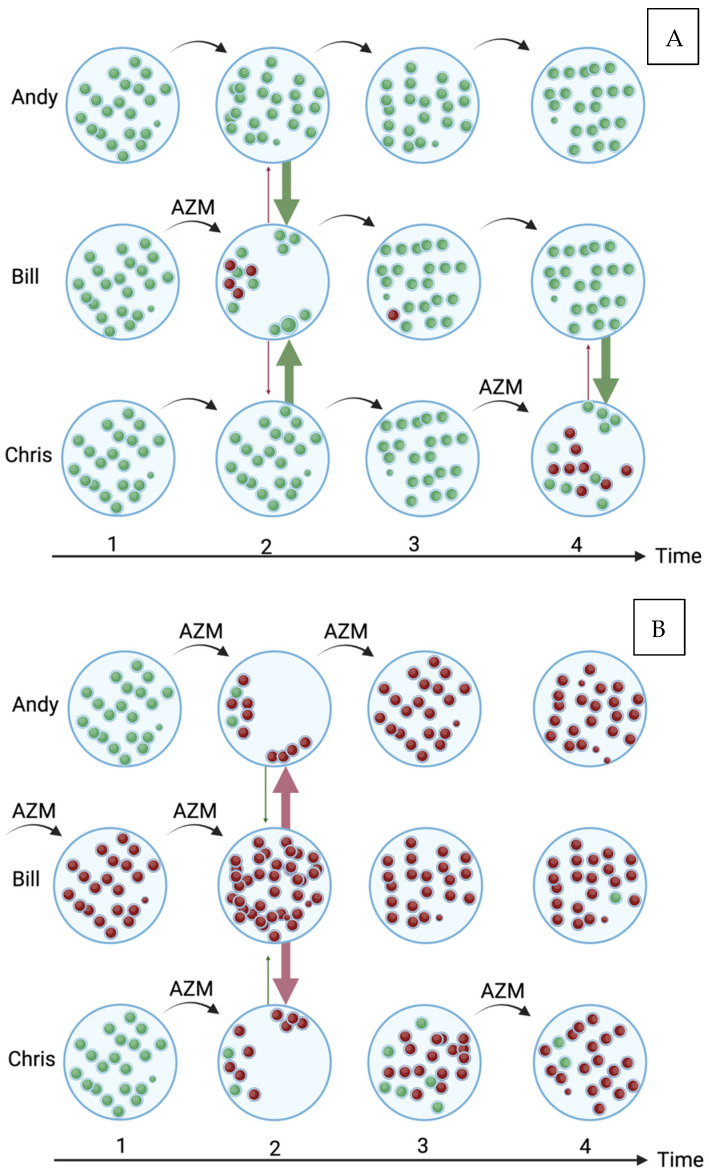
A schematic of how high antimicrobial consumption in a population could obscure the association between azithromycin (AZM) consumption and azithromycin resistance in oral streptococci in three individuals: Andy, Bill and Chris. In the low-azithromycin-consumption population (**A**), all the streptococci are susceptible (green) to azithromycin at baseline. Bill takes azithromycin at time point 1, which has the twin effect of selecting for resistant streptococci (red) and reducing the abundance of susceptible streptococci [27]. This niche can be refilled by the growth of endogenous streptococci but also via transmission from Andy and Chris [27]. If the resistant streptococci have a fitness cost, then the susceptible streptococci should replace the resistant isolates. Azithromycin administered at the third time point to any individual would still have a detectable effect on resistance. In the high-consumption population (**B**), Andy, Bill and Chris take azithromycin at time point 1, which has the same effect, with a notable difference in that due to Bill’s preceding azithromycin exposure, the main effect of the azithromycin is to increase the absolute number of resistant streptococci in his oropharynx [27]. This in turn, increases the probability that the transmission of his resistant streptococci to Andy and Bill will take place to fill their oral niches that have been depopulated through azithromycin (red arrows). Continuing exposure to azithromycin leads to all three individuals being colonised predominantly by azithromycin resistant streptococci. In this setting, from time point 3 onward, there would be a high probability of finding no association between azithromycin consumption and resistance. (For a review of all the five population-level pathways that link antimicrobial exposure to AMR, please see [27]; Created with BioRender.com).

**Table 1 antibiotics-13-00188-t001:** Demographics, antimicrobial read abundance and MICs of *Neisseria subflava* in the three study populations.

	MSM ABs (Median (IQR))	MSM No Abs [N (%)]	General Population [N (%)]
Men	32 (100)	32 (100)	10 (31.3)
Age categories (N persons per category (%)			
20–29	5 (15.6%)	4 (12.5%)	5 (15.6%)
30–39	16 (50.0%)	10 (31.3%)	9 (28.1%)
40–49	6 (18.8%)	8 (25.0%)	9 (28.1%)
50–59	4 (12.5%)	7 (21.9%)	8 (25.0%)
60–69	1 (3.1%)	2 (6.3%)	1 (3.1%)
70–79	0 (0%)	1 (3.1%)	0 (0%)
Antibiotic exposure in the previous 6 months, *n* (%)			
B-lactams	25 (78.1%)		
Macrolides	19 (59.4%)		
Fluoroquinolones	2 (6.3%)		
Others	8 (25.0%)		
*N. subflava* median MIC (ug/mL) #			
Ceftriaxone	0.035 (0.023–0.047)	0.032 (0.023–0.06)	0.036 (0.027–0.056)
Azithromycin	3 (1.75–4)	1.75 (0.38–6)	3 (2–4)
Ciprofloxacin	0.03 (0.01–0.38)	0.02 (0.01–0.192)	0.04 (0.02–0.19)
Antimicrobial read abundance (normalised read count)			
Macrolides	99 (0–539)	0 (0–351)	52 (0–167)
Betalactams	0 (0–48)	0 (0–65)	0 (0–66)
Fluoroquinolones	0 (0–83)	0 (0–101)	0 (0–63)
Tetracyclines	0 (0–527)	0 (0–415)	0 (0–134)

Notes: # The European Committee on Antimicrobial Susceptibility Testing (EUCAST) does not provide breakpoints for *N. subflava*. The breakpoints for antimicrobial resistance in *N. gonorrhoeae* are as follows: ceftriaxone > 0.125 mg/L, azithromycin epidemiological cutoff (ECOFF) = 1 mg/L; ciprofloxacin > 0.06 mg/L (https://mic.eucast.org/, accessed on 12 February 2024). AB—antibiotics; MSM—men who have sex with men.

**Table 2 antibiotics-13-00188-t002:** Linear regression of association between *Neisseria subflava* MICs for azithromycin and ceftriaxone (log values) and consumption of macrolides and beta-lactams, respectively.

	Coef. (95% CI)	*p*-Value
Azithromycin/macrolides	0.19 (−1.06–1.45)	0.759
Ceftriaxone/betalactams	−0.03 (−0.66–0.37)	0.926

**Table 3 antibiotics-13-00188-t003:** Zero-inflated negative binomial regression of association between the abundances of macrolide, beta-lactam, fluoroquinolone and tetracycline resistance-associated genes and the consumptions of these classes of antimicrobials.

Antimicrobial Consumption/Gene Abundance	Coef. (95% CI)	*p*-Value
Azithromycin/macrolides	0.23 (−0.30–0.77)	0.393
Ceftriaxone/betalactams	−0.03 (−0.66–0.37)	0.926
Fluoroquinolone/fluoroquinolone	−0.35 (−1.67–0.96)	0.598
Tetracycline/tetracycline	0.03 (−0.60–0.67)	0.916

## Data Availability

The data used in this study is available from the authors on reasonable request.

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
