# Peer review of "Lack of Association between Antimicrobial Consumption and Antimicrobial Resistance in a HIV Preexposure Prophylaxis Population: A Cross-Sectional Study"

_antibiotics, 2024, doi:10.3390/antibiotics13020188_

Round 1
Reviewer 1 Report
Comments and Suggestions for Authors
Thank you for your submission.
The paragraph form line 220-244 can be clearer with a focus on how this study can be improved to potentially detect a difference.
I agree with the sample size being small. Is there a target sample size that would have been better? Thirty is the absolute minimum for any sort of statistical analysis.
Line 226, starting with "Although studies..." When you reference multiple studies in your discussion, please include more than one study in the reference.
I also agree with the limitation of including only 1 Neisseria species may have limited identifying resistance. What other species should be considered and based on what evidence.
One aspect not seen in the results or discussion is the degree of the antibiotic exposure. One course of antibiotics is much different than 10 exposures. Duration of antibiotic use is also a potential confounder.
Line 246, a number of studies have found... Reference #10 and #20.
Reference 10, does not state data results from the PrEP cohort. It uses consumption data from various countries and antimicrobial resistance rates. They only theorize impact on PrEP as an example. The study is also not a RCT and can not prove causation but only correlation. Off topic, issues with comparing resistance rates from various countries is not just antimicrobial selective pressure but is more complex due to varying healthcare systems, access to testing, and antibiotics. Low/middle income countries may difference levels of access to antibiotics (prescription vs over-the-counter), but more variable access to culturing and susceptibility testing due to resource limitations.
So, reference #10 should not be used for the support of PrEP cohort using more antibiotics. Reference #20 is a better example.
Are the paragraphs from 267 to 285 really necessary? The paragraphs seems to deviate from the original study goals. A more condensed version might be more appropriate to keep the focus on antimicrobial exposure and resistance.
Reviewer 2 Report
Comments and Suggestions for Authors
The title is poorly worded. It should not contain a denial word such as “no” and the use of any abbreviation (in this case PrEP) should be avoided unless it is a universal one (e.g., WHO, HIV or PCR). The organism studied (Neisseria subflava) was also not mentioned.
PrEP was not spelt out in the abstract.
Why was N. gonorrhoeae listed as a keyword when the actual Neisseria species studied was N. subflava?
Why was Neisseria subflava chosen as the study organism? Compared to N. gonorrhoeae, it is generally considered a lowly pathogenic human commensal. Thus, its selection as the representative pathogen in PrEP patients has to be clearly justified.
It is difficult to understand the data presented in tables. For example, in Table 1, the line for “men” contains data such as 32 (100) and 32 (31.3). Do the numbers in brackets represent the median (IQR) as specifically stated in the heading above it? Also, tables should have a footer to spell out the abbreviations used (in this case, MSM and AB).
In Table 1, there should be interpretation of the MICs presented as well. For example, is an azithromycin MIC of 3 ug/mL susceptible?
The discussion should not contain self-incriminatory headings such as “We may have missed an association...” I fact, a discussion section without headings is better for this manuscript.
Reviewer 3 Report
Comments and Suggestions for Authors
The present study conducted in general and 2 cohorts of MSM (men who have sex with men) populations reports the presence or absence of any association between antimicrobial consumption and resistance by using genotypic and phenotypic markers. The study is well conducted, the methodology is clearly described and the manuscript is crisply written. No major comments from my side except a small suggestion of adding conclusion at the end, which should in-fact highlight the fact that absence of any association between AMU and AMR as observed in the current study should not be extrapolated to individuals with low levels of exposure. Conclusion in the abstract may also be re-phrased accordingly.
Comments on the Quality of English LanguageOverall fine; minor editing may be required.
